# Oral Supplementation with the Polyamine Spermidine Affects Hepatic but Not Pulmonary Lipid Metabolism in Lean but Not Obese Mice

**DOI:** 10.3390/nu14204318

**Published:** 2022-10-15

**Authors:** Sophia Pankoke, Christiane Pfarrer, Silke Glage, Christian Mühlfeld, Julia Schipke

**Affiliations:** 1Institute for Functional and Applied Anatomy, Hannover Medical School, 30625 Hannover, Germany; 2Biomedical Research in Endstage and Obstructive Lung Disease Hannover (BREATH), German Center for Lung Research, 30625 Hannover, Germany; 3Institute for Anatomy, University of Veterinary Medicine, Foundation, 30173 Hannover, Germany; 4Institution for Laboratory Animal Science, Hannover Medical School, 30625 Hannover, Germany

**Keywords:** diet-induced obesity, spermidine, caloric restriction mimetic, lung lipid metabolism, liver lipid metabolism, high fat diet, high sucrose diet

## Abstract

The polyamine spermidine is discussed as a caloric restriction mimetic and therapeutic option for obesity and related comorbidities. This study tested oral spermidine supplementation with regard to the systemic, hepatic and pulmonary lipid metabolism under different diet conditions. Male C57BL/6 mice were fed a purified control (CD), high sucrose (HSD) or high fat (HFD) diet with (-S) or without spermidine for 30 weeks. In CD-fed mice, spermidine decreased body and adipose tissue weights and reduced hepatic lipid content. The HSD induced hepatic lipid synthesis and accumulation and hypercholesterolemia. This was not affected by spermidine supplementation, but body weight and blood glucose were lower in HSD-S compared to HSD. HFD-fed mice showed higher body and fat depot weights, prediabetes, hypercholesterolemia and severe liver steatosis, which were not altered by spermidine. Within the liver, spermidine diminished hepatic expression of lipogenic transcription factors SREBF1 and 2 under HSD and HFD and affected the expression of other lipid-related enzymes. In contrast, diet and spermidine exerted only minor effects on pulmonary parameters. Thus, oral spermidine supplementation affects lipid metabolism in a diet-dependent manner, with significant reductions in body fat and weight under physiological nutrition and positive effects on weight and blood glucose under high sucrose intake, but no impact on dietary fat-related parameters.

## 1. Introduction

Diet-induced metabolic diseases such as obesity and type 2 diabetes are global health problems, mainly caused by an imbalance of consumed and expended calories [1]. Not only excess consumption of fat, but also high sugar intake is associated with dyslipidemia, liver steatosis and insulin resistance [2]. Although it is known that dietary changes and physical activity are effective intervention strategies against obesity and its comorbidities, it is often very difficult for affected individuals to change their lifestyle in a sufficient and permanent way [3]. Thus, supplementary treatments that support weight loss and/or alleviate the pathological consequences of unhealthy nutrition and obesity are needed. The naturally occurring molecule spermidine holds potential for alleviating diet-induced pathological effects at various levels [4].

The diamine putrescine and the polyamines spermidine and spermine are polycations, which are associated with a variety of cellular processes including transcription, translation and membrane stability [5,6]. The intracellular polyamine pools are tightly controlled by a complex regulatory system involving biosynthesis and catabolism, as well as uptake from extracellular sources [7]. For polyamine biosynthesis, ornithine is converted into the diamine putrescine by ornithine decarboxylase. From putrescine, spermidine and spermine are synthesized by spermidine synthase and spermine synthase, respectively, by utilizing decarboxylated S-adenosyl methionine [5,6]. Degradation and secretion of spermidine and spermine requires acetyl-CoA-dependent acetylation by spermidine/spermine N(1)-acetyltransferase (SSAT) [5]. The acetylated polyamines are then either excreted from the cell or oxidized to putrescine or spermidine by polyamine oxidase [5,8]. Besides the endogenous synthesis, the consumption of polyamine-rich foods such as wheat germ or soybean products is suited to increase polyamine levels in the body [9,10]. While tissue concentrations of spermidine decline with age, the supplementation with this polyamine results in an increased lifespan and cardioprotective effects in mice and other model organisms [11,12].

Polyamines are associated with the cellular glucose and lipid metabolism [4]. The overexpression of SSAT in transgenic mouse models leads to an accelerated polyamine flux and enhanced consumption of acetyl-CoA. This in turn results in a reduced fatty acid synthesis, a diminished lipid accumulation, an increased glucose and palmitate oxidation and an elevated energy expenditure, ultimately leading to a distinct lean phenotype even under a high fat diet regimen [13,14,15,16]. Mice fed a polyamine-enriched chow have similar body weights as control diet-fed animals, even though they consume more food [17], and oral spermidine supplementation in combination with physical activity significantly lowers the body weights of mice fed a high sucrose diet [18]. Moreover, the polyamine spermidine is discussed as a caloric restriction mimetic (CRM) candidate because of its capacity to induce autophagy and deplete acetyl-CoA. CRMs are pharmacological substances that mimic some effects of caloric restriction without the need to reduce calorie intake [19].

The liver plays a central role in the majority of metabolic processes and is responsible for lipid storage and distribution. Consequently, hepatic alterations such as liver steatosis and fibrosis are frequent obesity-related pathologies that are associated with comorbidities such as non-alcoholic fatty liver disease (NAFLD) or non-alcoholic steatohepatitis (NASH) [20,21]. In rodents, oral administration of polyamines is able to promote liver regeneration after ischemia and reperfusion injury and protects the liver from induced fibrosis [22,23]. Under a high fat diet regimen, a daily intraperitoneal dose of spermine improves glucose utilization and decreases body weights of obese mice, whereas a daily injection of spermidine even led to reduced hepatic steatosis [24,25]. In humans, it was shown that NASH parameters are negatively correlated with spermidine intake and serum levels, whereas in mice the oral supplementation of spermidine was able to ameliorate NASH by regulating the liver lipid metabolism [26]. Furthermore, increased levels of polyamines in white adipose tissue and the liver might confer resistance to diet-induced obesity and NAFLD [27]. 

Single studies indicate that liver and lung pathologies are linked to each other. Patients with liver diseases are at higher risk for ARDS, and a case study demonstrated that liver transplantation was efficient in resolving ARDS, implying that a healthy liver protects the lung from injury [28]. Hepatic steatosis accompanies alveolar proteinosis, indicating close links between the pulmonary and the hepatic lipid metabolism [29]. The lung, especially the alveolar area, maintains an active lipid metabolism for the biosynthesis of surfactant. Surfactant is essential for normal lung function and consists mainly (~90%) of lipids [30]. Only scarce knowledge exists about polyamine-related effects on the lung. Alveolar epithelial cells are equipped with a distinct polyamine uptake system and actively accumulate spermidine upon stress triggers such as hyperoxia, hypoxia and ozone exposure [31,32,33,34,35]. However, the functional consequences of this spermidine accumulation remain to be elucidated. In idiopathic pulmonary fibrosis, pulmonary spermidine levels are reduced, and spermidine administration elicits beneficial effects on bleomycin-induced experimental lung fibrosis [36,37]. In obese mice, oral spermidine supplementation efficiently alleviates obesity-related thickening of the endothelium and the air–blood barrier and significantly increases the endothelial surface area, pointing to beneficial effects on gas exchange in the fatty lung [38]. 

Taken together, these findings indicate a therapeutic potential for spermidine against obesity-related pathological alterations in a systemic, hepatic and pulmonary context. However, several open questions remain. Previous studies often analyzed genetic modifications and intraperitoneal injection of polyamines, which are hardly applicable for patients. Thus, more insight is needed into the oral administration of spermidine. Only limited knowledge exists on whether spermidine supplementation interferes with lipid metabolism, as was shown for SSAT overexpression, and whether spermidine-related effects depend on the underlying nutritional status and differ between organ systems. Thus, this study examined the effects of oral spermidine supplementation for 30 weeks under control diet, high sucrose diet and high fat diet conditions in male mice. Body weight; circulating lipid concentrations; lipid deposition in fat depots, liver and lung; hepatic and pulmonary expression of lipid metabolism-related enzymes; and glucose homeostasis-related parameters were measured to assess metabolic consequences of the different conditions.

## 2. Materials and Methods

### 2.1. Animal Studies

Male C57BL/6N mice were purchased from Charles River (Sulzfeld, Germany) at an age of 5 weeks and housed individually in cages equipped with shelters and nesting material under temperature-controlled conditions (21 ± 2 °C). The mice had ad libitum access to food and water. After 1 week of acclimatization, mice were randomly allocated to one of three diet groups: (1) a purified control diet (CD; S3542-E040, ssniff Spezialdiäten, Soest, Germany), (2) a purified high sucrose diet (HSD; S3542-E042, ssniff) or (3) a purified high fat diet (HFD; S3542-E044, ssniff). The composition of the diets is given in Table 1. Half of each diet group received drinking water containing 3 mM spermidine (pH 7.4, 16 mosm/kg; Sigma-Aldrich, St. Louis, MO, USA). The spermidine-containing water was replaced every 3–4 days and was kept protected from light. This administration protocol was shown previously to enhance spermidine concentrations in blood and lung tissue [12,38]. Body weights and food consumption were measured regularly.

This study was performed in accordance with the European Directive 2010/63/EU and with German animal protection laws. The Lower Saxony State Office for Consumer Protection (LAVES; file number 18/2841) approved and permitted all animal experiments.

### 2.2. Blood Plasma Analysis

In weeks 10, 20 and 29, mice were feed-deprived for 6 h, retro-orbital blood was collected and blood plasma was isolated by centrifugation. Plasma concentrations of circulating glucose, lipids and transaminases were assessed in the Institute for Clinical Chemistry, Hannover Medical School, using automated analyzer systems and kits (Roche Diagnostics, Mannheim, Germany). Insulin levels were analyzed in duplicates with an ultrasensitive mouse insulin ELISA (#90080, Chrystal Chem, Elk Grove Village, IL, USA). The homeostasis model assessment for insulin resistance (HOMA-IR) and 1/fasting insulin as a surrogate index for insulin resistance were calculated and normalized to body weight as described before [39]. HOMA-IR and 1/fasting insulin were normalized to body weight since this was shown to considerably enhance the correlation with glucose clamp-derived measures [40].

### 2.3. Oral Glucose Tolerance Test

In week 26, mice were feed-deprived for 6 h. A baseline blood sample was collected from the tail vein. Then, 1 mg glucose/g body weight was administered by oral gavage. Tail vein blood samples were obtained after 15, 30, 60 and 120 min and analyzed by a glucometer (Wellion Calla; Med Trust, Ottendorf-Okrilla, Germany). The area under the glucose tolerance curve (AUC) was calculated using the trapezoidal rule.

### 2.4. Necropsy and Organ Preparation

After 30 weeks, mice were killed under deep anesthesia (ketamine 100 mg/kg body weight, xylazine 5 mg/kg body weight, i.p.). The right lung lobes were ligated, snap-frozen in liquid nitrogen and stored at −80 °C for mRNA expression analyses. Left lungs were fixed by tracheal instillation of 1.5% paraformaldehyde/1.5% glutaraldehyde/0.15 M HEPES buffer at a hydrostatic pressure of 20 cm H_2_O, incubated in fixative solution for at least 24 h and further processed according to design-based stereological methods [41]. Briefly, left lung volumes were determined by Archimedes’ principle [42], and lung sampling was performed according to systematic uniform random sampling methods [43]. Tissue blocks were embedded in epoxy resin (Epon; Serva, Heidelberg, Germany) as described elsewhere [39] and analyzed by transmission electron microscopy.

Livers were isolated, weighed and cut into ~1 mm slices. Slices were randomly allocated to snap freezing and storage at −80 °C for mRNA expression analyses, or fixation in 4% paraformaldehyde/0.2 M HEPES buffer (pH 7.4) and embedding in epoxy resin as described elsewhere [39] for histological analysis. 

Epididymal, retroperitoneal and interscapular fat depots were isolated, weighed and snap-frozen in liquid nitrogen and stored at −80 °C.

The experimental design is illustrated in Figure 1.

### 2.5. Lipid Droplet Quantification

Due to the smaller size of septal lipid droplets (LDs), lipid quantification in the lung was performed at the electron microscopy level, whereas light microscopy analysis was sufficient for liver LD assessment.

For LD analysis in the lung, ~60 nm ultrathin sections from three randomly chosen epoxy resin blocks per animal were cut and mounted on copper grids. A transmission electron microscope (FEI Morgagni; Eindhoven, Netherlands) with a digital camera (Veleta; Olympus Soft Imaging Solutions, Münster, Germany) was used to obtain ~50 random fields of view per section at a primary magnification of 14,000×. The volume of septal lipid droplets was assessed by point probes utilizing the STEPanizer stereology tool [44,45]. Points hitting LDs (P(LD)) and points hitting the lung septa representing the reference space (P(sept)) were counted for the estimation of the volume fraction (V_V_) of septal LDs: $$V_{V}\left(\mathrm{LD}/\mathrm{sept}\right)=\sum P\left(\mathrm{LD}\right)\;/\sum P\left(\mathrm{sept}\right)$$

The V_V_(LD/sept) was multiplied by 100 to calculate the LD percentage.

For LD analysis in the liver, 1 µm thick sections from three randomly chosen epoxy resin blocks per animal were cut, mounted on glass slides and stained with toluidine blue. Whole sections were digitalized with an AxioScan.Z1 Scanner (Zeiss, Oberkochen, Germany) at a primary magnification of 40×. Subsequently, automated image analysis was performed using the Tissuemorph DP software from Visiopharm (version 5.3.1.1723, Hoersholm, Denmark) as described before [39].

### 2.6. mRNA Expression Analysis

mRNA isolation from liver and lung tissue was performed using the NucleoSpin RNA/Protein Kit (#740933.250, Marcherey-Nagel, Düren, Germany) according to the manufacturer’s manual. RNA concentration was measured with a NanoDrop 2000 Spectrophotometer (Thermo Fisher Scientific, Waltham, MA, USA). cDNA was synthesized from 1 µg mRNA using the iScript cDNA Synthesis Kit (BioRad, Hercules, CA, USA). Real-time PCR analysis was performed using a C1000 Thermal Cycler (CFX384 Real-Time System, BioRad) as described before [39]. Gene expression analysis was conducted in triplicates, and the primers used are listed in Table 2. The relative mRNA expression was assessed by normalization to the housekeeping gene HPRT (ΔCt) and CD-fed animals (ΔΔCt) and was calculated as 2^ΔΔCt^.

### 2.7. Statistics

Two-way repeated-measures ANOVA (2W-RM-ANOVA) and two-way ANOVA (2W-ANOVA), each followed by a post hoc Tukey test, were performed as indicated in figure legends using Sigma Plot version 13.0 (Systat Software Inc., San Jose, CA, USA). *p* < 0.05 values were considered statistically significant and are shown in the figures as stated in the figure legends. Data are either expressed as values of individual mice with means indicated by horizontal bars (dot plots) or expressed as group means with standard derivation (bars).

## 3. Results

HSD-fed mice ingested significantly more calories (Figure 2F,J) but showed similar body weights (Figure 2A,E) and fat depot weights (Figure 2K–M) when compared to CD. The HFD resulted in the highest calorie intake (Figure 2F,J) and highest body weights (Figure 2A,E) of all diet groups. Additionally, the weight of epididymal and retroperitoneal white adipose tissue (WAT) and the weight of the interscapular fat depot consisting of WAT and brown adipose tissue (BAT) were significantly elevated in the HFD group (Figure 2K–M).

A spermidine supplementation of CD- and HSD-fed animals resulted in diminished body weights at the end of the experimental duration (Figure 2B,C,E) and a slightly reduced calorie intake between weeks 26 and 30 (Figure 2G,H,J). Only in the CD group, this was accompanied by a reduction in WAT after 30 weeks (Figure 2K,L). In contrast, spermidine had no impact on HFD-related parameters (Figure 2).

The high sucrose consumption had no impact on glucose homeostasis compared to control conditions (Figure 3). In contrast, excess fat intake led to a delayed glucose clearance in glucose tolerance testing (Figure 3A), higher plasma insulin concentrations (Figure 3G), an increased insulin resistance (Figure 3H) and a diminished insulin sensitivity (Figure 3I) indicating a prediabetic state in HFD-fed mice. 

Spermidine lowered glucose levels in CD (Figure 3B) and HSD (Figure 3F) groups but did not affect any of the HFD-induced glucose metabolism alterations (Figure 3D,G–I).

Both high sucrose and high fat intake resulted in significantly elevated total cholesterol, high-density lipoprotein (HDL) cholesterol and low-density lipoprotein (LDL) cholesterol plasma concentrations (Figure 4A,E,I). In contrast, only the HFD increased transaminase levels (ALT after 29 weeks, Figure 4Q; AST after 20 weeks, Figure 4U), indicating liver injury. None of the diets affected plasma triglycerides (Figure 4M).

Spermidine supplementation showed reducing effects on triglycerides and ALT after 29 weeks in HSD-fed mice (Figure 4O,S). In the HFD group, spermidine increased HDL and LDL cholesterol concentrations (Figure 4H,L).

These diet- and spermidine-induced systemic changes were associated with distinct hepatic and pulmonary expression profiles of lipid metabolism-related enzymes. Expression of transcription factors sterol regulatory element binding transcription factor 1 (SREBF1), SREBF2 and peroxisome proliferator-activated receptor α (PPARA) was induced only in the liver, and only under HFD feeding (Figure 5A). Hepatic expression levels of the lipogenic SREBF1 and the cholesterogenic SREBF2 were significantly reduced by spermidine supplementation in HSD- and HFD-fed mice (Figure 5A). Enzymes related to fatty acid and triglyceride synthesis were expressed at significantly higher levels upon high sucrose intake in the liver, but not in the lung (Figure 5C–E,G). This was partly alleviated by spermidine supplementation (Figure 5C,E). Within the lung, the HFD reduced the expression of the G3P acyltransferase (GPAM) (Figure 5G).

Regarding lipid catabolism, the experimental diets elicited differential effects in the liver. HFD induced adipose triglyceride lipase (ATGL), an enzyme that catalyzes the hydrolysis of triglycerides to diacylglycerol and thereby initiates lipolysis, as well as carnitine palmitoyltransferase 1 (CPT1A), which is located at the outer mitochondrial membrane and is implicated in β-oxidation of fatty acids (Figure 5F,I). In contrast, HSD induced the expression of both subunits of the mitochondrial trifunctional protein (HADHA, HADHB), a protein present at the inner mitochondrial membrane catalyzing three out of four reactions in mitochondrial β-oxidation (Figure 5I). Spermidine supplementation significantly reduced the HFD-related induction of ATGL and the HSD-related induction of HADHA and HADHB to control levels (Figure 5F,I). Within the lung, only minor effects on lipolysis were observed, including increased expression of ATGL in the HFD-S group and of CPT1A in the HFD group (Figure 5H,J).

The main amount of hepatic LDs was located in hepatocytes. The induction of lipogenic enzymes in the liver upon excess sucrose intake resulted in hepatic lipid accumulation (Figure 6B,C). In addition, the high fat intake in the HFD group resulted in liver steatosis, consistent with the observed higher plasma transaminase levels. Spermidine supplementation significantly reduced hepatic lipid amounts in the control diet group, in line with the lower body and WAT weights. These changes were not reflected by the liver weight (Figure 6A). Within the lung septa, LDs were mainly located in interstitial fibroblasts and occasionally in epithelial or endothelial cells. Neither diet nor spermidine supplementation had an impact on the lipid fraction within these cells (Figure 6E,F). The lung volume was slightly elevated in HSD-S in comparison to HSD (Figure 6D).

## 4. Discussion

The naturally occurring polyamine spermidine is discussed as a CRM and therapeutic option for obesity and its related comorbidities and might elicit beneficial effects on hepatic and pulmonary disorders. In this study, oral spermidine supplementation was tested as an intervention strategy against diet-related alterations of the systemic, hepatic and pulmonary lipid metabolism.

Under control diet conditions, spermidine effectively reduced body and WAT weights and decreased liver lipid content. These findings are in accordance with studies on SSAT overexpression that resulted in lower body weights and diminished epididymal WAT in genetically modified mice [16,46]. The rationale behind this observation is that the increased expression of the polyamine catabolic enzyme SSAT results in an accelerated polyamine flux, thereby depleting acetyl-CoA. This subsequently leads to a diminished fatty acid synthesis and lipid accumulation, as well as an increased glucose and fatty acid oxidation and an elevated energy expenditure, ultimately leading to a lean phenotype [13,14,15]. Our data show that not only genetic modification of SSAT, but also oral intake of spermidine alters lipid deposition as well as body weight, but only under a physiological diet. Upon high sucrose intake, only body weight but not fat deposition was altered by spermidine supplementation; upon high fat intake, neither weight nor lipid accumulation was altered by spermidine supplementation. This is in contrast to reported genetic modifications of polyamine flux via SSAT, which prevented HFD-related lipid accumulation and body weight increase [4,46,47]. This indicates that by oral supplementation a distinct polyamine concentration within the body is reached that is adequate to affect lipid metabolic processes under physiological conditions. However, these concentrations might be not sufficient if the hepatic lipid homeostasis is additionally stimulated by dietary carbohydrates or fat. This notion is supported by the reduced hepatic expression of lipogenic and cholesterogenic transcription factors SREBF1 and SREBF2 in HSD-S and HFD-S groups, albeit without major further consequences on lipogenic enzyme expression or liver lipid content (Figure 5 and Figure 6).

To use spermidine as a therapeutic approach against obesity-related morbid consequences, oral supplementation as an administration route applicable for patients should be effective. Studies exploring oral supplementation of spermidine are scarce. Ni et al. reported that oral administration of spermidine did not reduce body weights of Western diet-fed animals, but, in contrast, improved serum triglyceride (TG) levels, hepatic steatosis, glucose tolerance and insulin resistance [26]. In our study, serum TG levels were decreased by spermidine in the HSD-S group in week 29 as well as in the HFD-S group in week 10 (Figure 4O,P), pointing to diet- and time-dependent spermidine effects. In contrast, hepatic steatosis, glucose tolerance and insulin resistance were not affected in our experimental setup (Figure 3 and Figure 6). One obvious difference between the study of Ni et al. and our study is the different diet systems. While we used either an HFD (35% fat, mainly cholesterol-rich lard) or an HSD (46% sucrose), Ni et al. used a Western diet composed of 40% fat, 41% sucrose and 1.25% cholesterol and additionally used fructose/glucose-enriched drinking water [26]. Therefore, the diet-related metabolic profiles of the mice were likely different. Moreover, Ni et al. utilized another mouse substrain (C57BL/6J vs. C57BL/6N in our study). It was shown previously that C57BL/6J mice harbor an Nnt gene deletion that results in impaired glucose tolerance [48], presumably contributing to the differing observations. Another important difference is also the used CD, namely a purified diet matching the experimental diets in this study and conventional chow in the study of Ni et al. Grain-based chow consists of natural ingredients that vary between charges due to harvest supply [49] and contains various concentrations of phytoestrogens possibly influencing study results [50,51]. This restricts its application as control food for diet-induced obesity studies and highlights the importance of control diet composition for animal studies on diet-induced obesity. In the same line of evidence, spermidine showed no systemic effects in mice fed a conventional, starch-rich CD as previously published by our lab [18], in contrast to the body weight reduction reported here with a custom-made, fiber-rich CD. A direct comparison between these diets revealed a carbohydrate-related body weight increase and liver lipid accumulation [39] that possibly masked spermidine-related effects in the former study. This highlights the need for future studies on spermidine supplementation in the context of different nutrition regimens. Our result that spermidine reduces lipid deposition in lean mice might even indicate adverse effects in normal-weight or undernourished persons; thus, future experiments should also include caloric restriction.

In high sucrose-fed animals, spermidine supplementation significantly reduced body weight and blood glucose concentrations. This is in accordance with previous studies reporting weight-reducing effects of oral spermidine supplementation in combination with physical activity in HSD-fed mice [18] and a normalizing effect of intraperitoneally administered spermidine on diabetes-related hyperglycemia in rats [52]. Together, these results indicate beneficial systemic spermidine-related effects under sucrose-rich nutrition. In contrast, liver lipid accumulation was not significantly altered, which may be due to the fact that the expression of both lipogenic (FASN and GPAM) and lipolytic (mitochondrial trifunctional protein) enzymes was reduced in the liver. As a limitation of this study, energy expenditure was not measured. Thus, the authors cannot exclude that diet- and spermidine-related differences in energy expenditure contributed to the findings reported here.

The liver lipid content was more than tripled in response to the HSD (12.6 ± 8.4%) and even 5-fold higher in HFD-fed mice (HFD 17 ± 7.6%) compared to the CD (CD 3.5 ± 1.6%). Excess sucrose intake activated hepatic de novo lipogenesis for energy storage. In contrast, HFD-feeding upregulated lipogenic transcription factors SREBF1, SREBF2 and PPARA, but without consequences on the expression of lipogenic enzymes, at least after 30 weeks. We cannot exclude that hepatic lipogenesis took place at earlier times of the experiment and maybe contributed to the lipid accumulation in liver and adipose tissues in HFD-fed mice.

In the lung, we observed only minor diet- and spermidine-related effects on expression profiles and lipid accumulation. Within the septa, several cell types fulfill lipid-related tasks, i.e. alveolar epithelial type 2 (AT2) cells (production and modification of complex lipid species for surfactant synthesis), interstitial fibroblasts (lipid storage) and alveolar macrophages (lipid recycling and breakdown) [53,54,55]. Since individual cell types were not addressed in this study, cell type-specific alterations were possibly not detected. In mice, it was shown before that consumption of carbohydrates and fat influences the lipid metabolism of AT2 cells in a distinct way, affecting surfactant composition and function [56]. Similarly, in obese diabetic rats, AT2 cell morphology and surfactant protein expression are altered [57]. A fat-enriched diet induced capillary remodeling including an increased endothelial thickness and a decreased endothelial surface area [38], whereas high sucrose intake resulted in elastic fiber changes associated with diminished pulmonary elasticity [58]. This points to macronutrient-dependent obesity-related changes in the lung, as observed for other parameters mentioned above. Excess fat intake reduced pulmonary expression of the lipogenic enzyme GPAM and induced the lipolytic enzyme CPT1A, indicating adaptive responses to the increased systemic lipid levels. However, this had no impact on septal lipid contents. Future studies are needed to better understand diet-related pulmonary changes at the cellular level and in the context of other organs. 

Rat and human alveolar epithelial cells possess a particularly active uptake system for polyamines [31,59,60]. Since these studies used isolated lung slices, a high dose of polyamines was supplied directly to the cells via the epithelial side. This is in contrast to our study in which the orally administered spermidine was taken up via the gut and was delivered with the blood, thus resulting in presumably lower concentrations arriving in the lung from the endothelial side. However, a previous study from our lab showed significant increases in pulmonary spermidine upon oral administration, confirming the bioavailability of dietary spermidine for the lung [38]. Moreover, spermidine significantly increased pulmonary expression of FASN and ATGL in HFD-fed mice (Figure 5), demonstrating subtle yet detectable diet-related spermidine effects within the lung. Polyamine accumulation within pulmonary cells is induced by exposure to hypoxia, hyperoxia or ozone [33,34,35], and spermidine supplementation significantly alleviates obesity-related thickening of the endothelium and the air–blood barrier [38]. This points to protective polyamine-related effects under stress conditions and warrants future studies on polyamines and the lung.

## 5. Conclusions

Taken together, our data show that oral administration of 3 mM spermidine is able to influence lipid metabolism, but its eventual effectiveness is dependent on the nutritional status of the individual. In control diet-fed, lean mice, spermidine reduced body weight and lipid contents in adipose tissues and the liver. Upon high sucrose intake, it still diminished body weight and blood glucose. However, spermidine supplementation of HFD-fed obese mice exerted no effects on body weight or lipid deposition. This indicates that spermidine supplementation is effective as a CRM, but only under specific diet-related conditions. The dose and dosage form of spermidine used here did not improve the metabolic profile of diet-induced obese mice; thus, this study does not support an unrestricted recommendation of spermidine as an obesity intervention strategy. 

## Figures and Tables

**Figure 1 nutrients-14-04318-f001:**
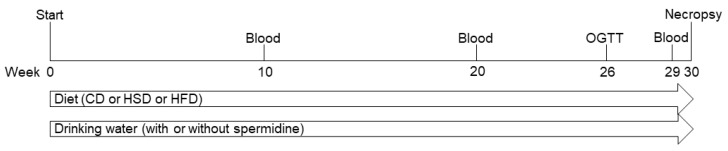
Experimental design. OGTT, oral glucose tolerance test; CD, control diet; HSD, high sucrose diet; HFD, high fat diet.

**Figure 2 nutrients-14-04318-f002:**
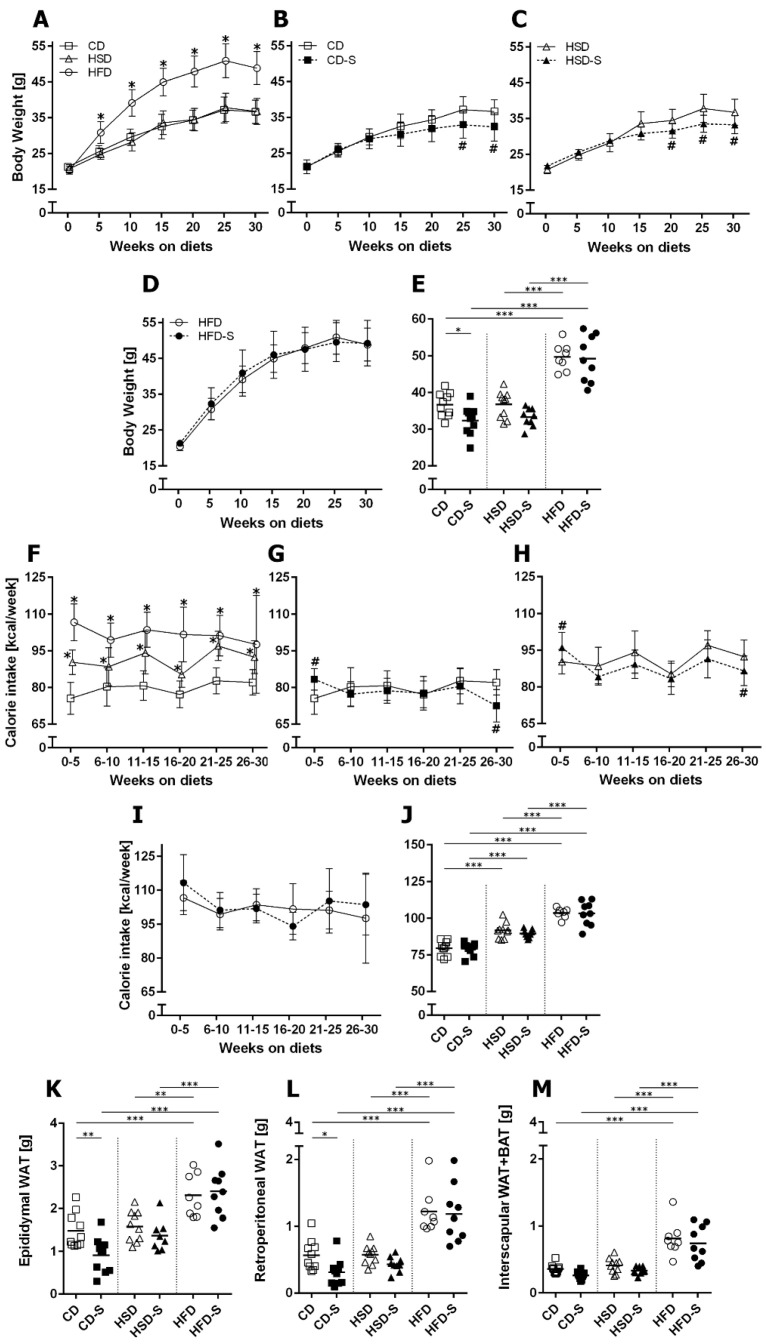
Effects of diets and spermidine on body weight, calorie intake and fat depots. (**A**) Diet-related effects on body weight gain. (**B**–**D**) Spermidine-related effects on body weight gain in respective diet groups. (**E**) Final body weights after 30 weeks. (**F**) Diet-related effects on calorie intake per week. (**G**–**I**) Spermidine-related effects on calorie intake per week in respective diet groups. (**J**) Mean calorie intake per week over 30 weeks. (**K**–**M**) Fat depot weights. (**A**–**D**,**F**–**I**) Values are group means ± SD; CD *n* = 10, CD-S *n* = 10, HSD *n* = 10, HSD-S *n* = 9, HFD *n* = 8, HFD-S *n* = 9; data were compared separately for diet and spermidine effects by 2W-RM-ANOVA followed by Tukey test; *p*-values < 0.05 are indicated by * for dietary effect or # for spermidine effect. (**E**,**J**–**L**) Data shown as values of individual mice, group means are indicated by horizontal lines; data were compared by 2W-ANOVA followed by Tukey test; * *p* < 0.05, ** *p* < 0.01, *** *p* < 0.001. CD, control diet; HSD, high sucrose diet; HFD, high fat diet; -S, spermidine.

**Figure 3 nutrients-14-04318-f003:**
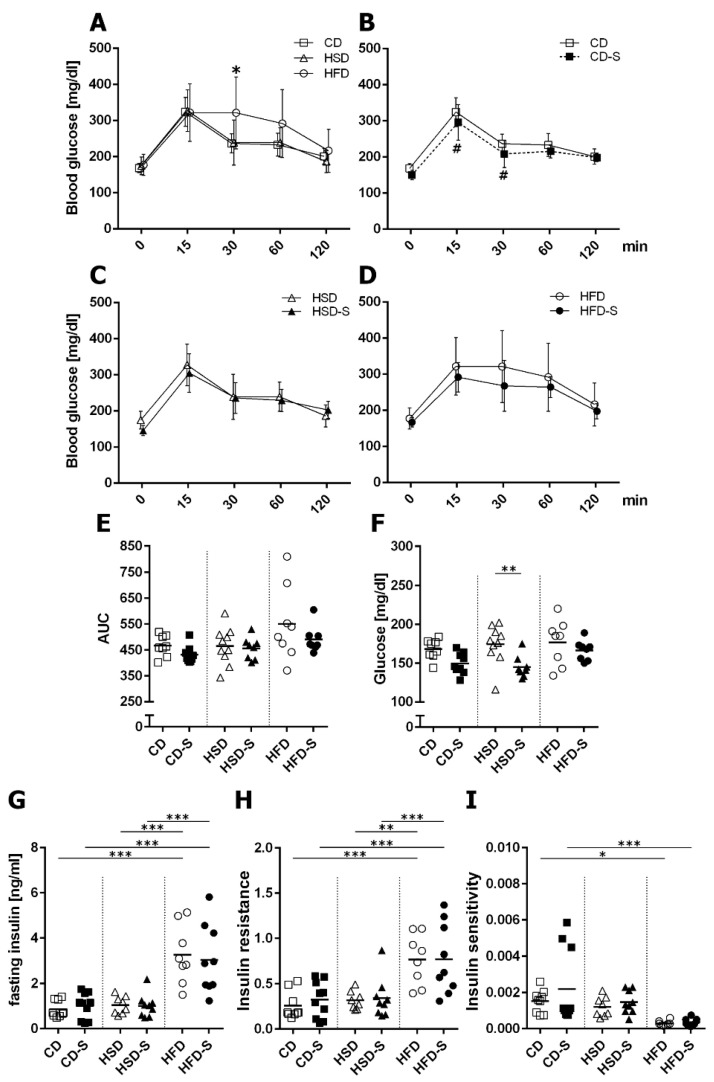
Effects of diets and spermidine on glucose homeostasis. (**A**) Diet-related effects on glucose tolerance. (**B**–**D**) Spermidine-related effects on glucose tolerance in respective diet groups. (**E**) Area under the curve (AUC) values for glucose tolerance curves. (**F**) Fasting plasma glucose concentrations. (**G**) Fasting plasma insulin concentrations. (**H**) HOMA insulin resistance related to body weight. (**I**) Insulin sensitivity measured by 1/insulin related to body weight. (**A**–**D**) Values are group means ± SD; CD *n* = 10, CD-S *n* = 10, HSD *n* = 10, HSD-S *n* = 9, HFD *n* = 8, HFD-S *n* = 9; data were compared separately for diet and spermidine effects by 2W-RM-ANOVA followed by Tukey test; *p*-values < 0.05 are indicated by * for dietary effect or # for spermidine effect. (**E**–**I**) Data shown as values of individual mice, group means are indicated by horizontal lines; data were compared by 2W-ANOVA followed by Tukey test; * *p* < 0.05, ** *p* < 0.01, *** *p* < 0.001. CD, control diet; HSD, high sucrose diet; HFD, high fat diet; -S, spermidine.

**Figure 4 nutrients-14-04318-f004:**
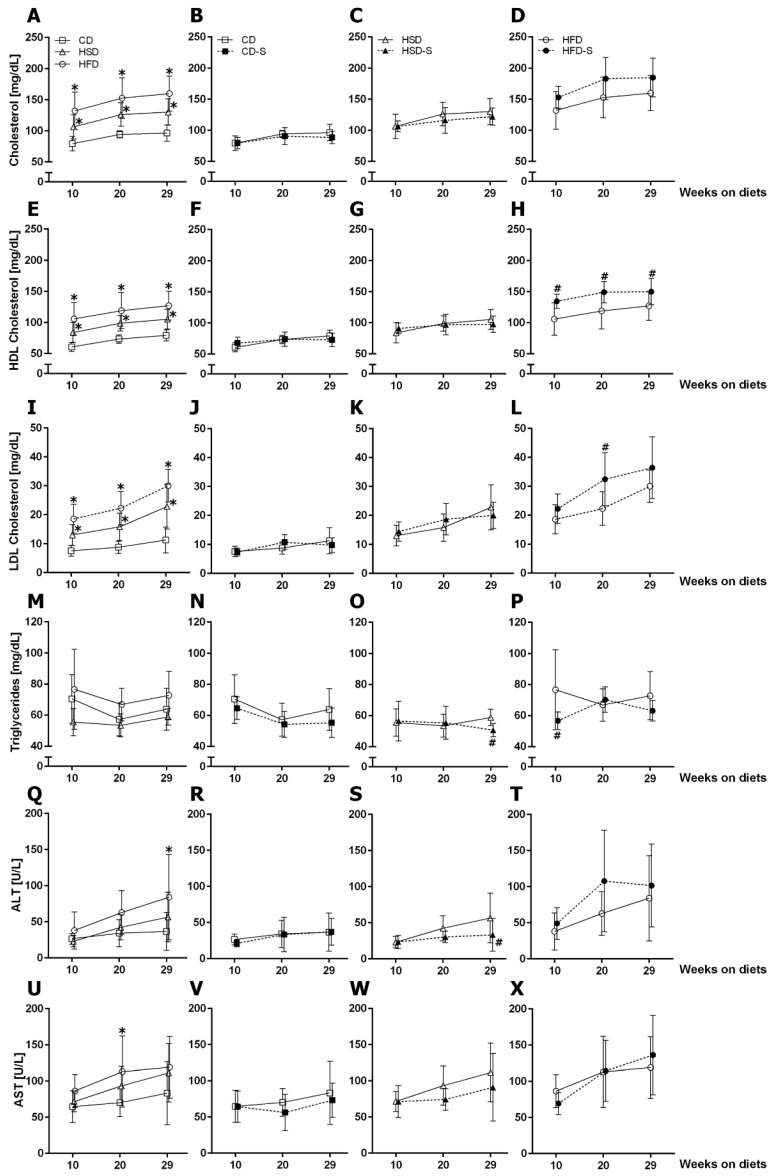
Effects of diets and spermidine on circulating lipids and enzymes. (**A**–**D**) Fasting plasma total cholesterol concentrations. (**E**–**H**) Fasting plasma HDL cholesterol concentrations. (**I**–**L**) Fasting plasma LDL cholesterol concentrations. (**M**–**P**) Fasting plasma triglyceride concentrations. (**Q**–**T**) Fasting plasma alanine transaminase concentrations. (**U**–**X**) Fasting plasma aspartate transaminase concentrations. Values are group means ± SD; CD *n* = 10, CD-S *n* = 10, HSD *n* = 10, HSD-S *n* = 9, HFD *n* = 8, HFD-S *n* = 9. Data were compared separately for diet and spermidine effects by 2W-RM-ANOVA followed by Tukey test; *p*-values < 0.05 are indicated by * for dietary effect or # for spermidine effect. CD, control diet; HSD, high sucrose diet; HFD, high fat diet; -S, spermidine.

**Figure 5 nutrients-14-04318-f005:**
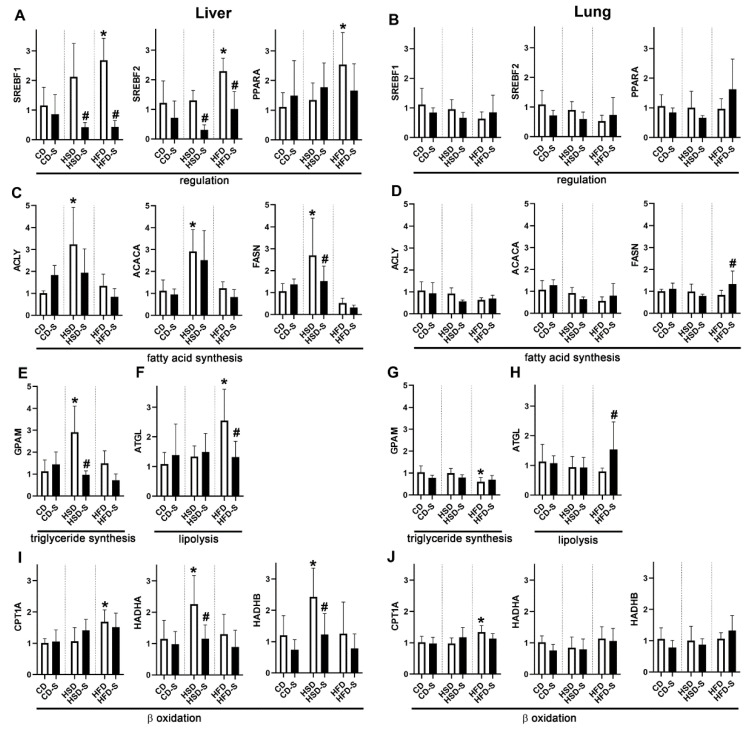
Effects of diets and spermidine on hepatic and pulmonary expression profiles of lipid metabolism-related proteins. (**A**,**B**) Sterol regulatory element binding transcription factors 1 and 2; peroxisome proliferator-activated receptor α. (**C**,**D**) ATP citrate lyase; acetyl-CoA carboxylase α; fatty acid synthase. (**E**,**G**) Glycerol-3-phosphate acyltransferase. (**F**,**H**) Adipose triglyceride lipase. (**I**,**J**) Carnitine palmitoyltransferase 1α; mitochondrial trifunctional protein, subunits α and β. Shown are fold expression values given as group means ± SD; *n* = 5 for every group. Data were compared by 2W-ANOVA followed by Tukey test; *p*-values < 0.05 are indicated by * for significant difference compared to CD (dietary effect) or # for significant difference compared to respective non-supplemented diet group (spermidine effect). CD, control diet; HSD, high sucrose diet; HFD, high fat diet; -S, spermidine.

**Figure 6 nutrients-14-04318-f006:**
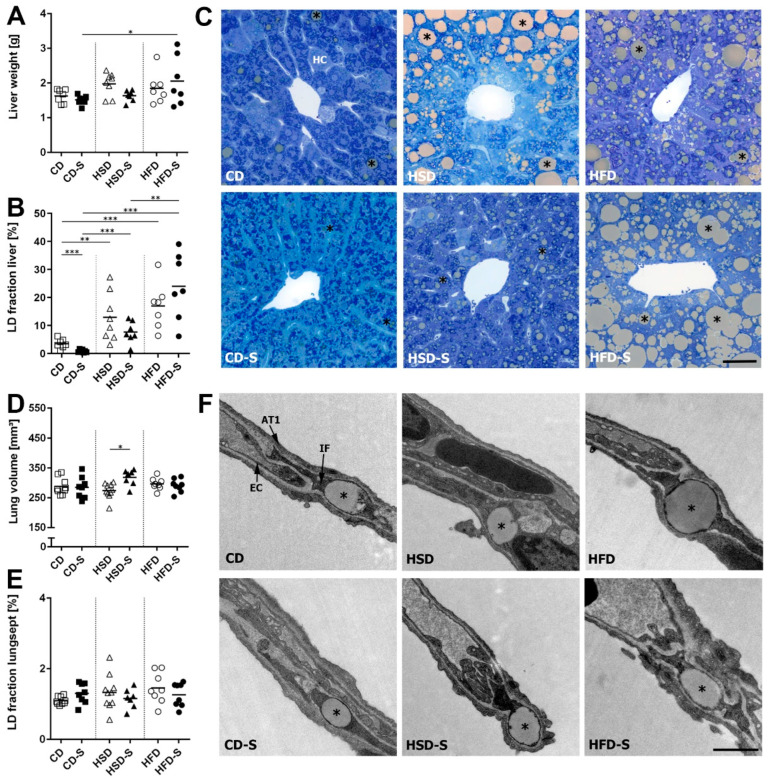
Effects of diets and spermidine on hepatic and pulmonary lipid accumulation. (**A**) Liver weight. (**B**) Percentage of area occupied by lipid droplets in the liver. (**C**) Representative light microscopy images of toluidine blue stained liver sections, scale bar = 50 µm. (**D**) Lung volume. (**E**) Percentage of area occupied by lipid droplets in alveolar septa. (**F**) Representative electron microscopy images of the pulmonary septal region, scale bar = 1 µm. (**A**,**B**,**D**,**E**) Data shown as values of individual mice, group means are indicated by horizontal lines; data were compared by 2W-ANOVA followed by Tukey test; * *p* < 0.05, ** *p* < 0.01, *** *p* < 0.001. CD, control diet; HSD, high sucrose diet; HFD, high fat diet; -S, spermidine; *, LD; HC, hepatocyte; IF, interstitial fibroblast; AT1, epithelial type 1 cell; EC, endothelial cell.

**Table 1 nutrients-14-04318-t001:** Composition of experimental diets.

Ingredient	CD	HSD	HFD
Fat (%)	4.1	4.1	34.8
Pork lard (%)	1.6	1.6	31.6
Soybean oil (%)	2.4	2.4	3.1
Carbohydrates (%)	68.5	70	31.8
Pectin/inulin (%)	13.6	0.2	0.2
Corn starch (%)	35.9	7.8	7.8
Maltodextrin (%)	11	11	11.4
Sucrose (%)	5.9	45.9	6.8
Protein (%)	17.6	17.6	23.7
Energy (kcal/g)	3.25	3.75	5.23
Kcal% Fat	11	10	60
Kcal% Carbohydrates	67	71	22
Kcal% Protein	22	19	18

CD, control diet; HSD, high sucrose diet; HFD, high fat diet.

**Table 2 nutrients-14-04318-t002:** Primers used for real-time PCR.

Target		BioRad Assay ID
SREBF1	Sterol regulatory element binding transcription factor 1	qMmuCIP0033121
SREBF2	Sterol regulatory element binding transcription factor 2	qMmuCIP0035147
PPARA	Peroxisome proliferator-activated receptor alpha	qMmuCEP0054952
ACLY	ATP citrate lyase	qMmuCEP0053217
ACACA	Acetyl-CoA carboxylase α	qMmuCIP0030034
FASN	Fatty acid synthase	qMmuCEP0054102
GPAM	Glycerol-3-phosphate acyltransferase	qMmuCIP0034231
ATGL	Adipose triglyceride lipase	qMmuCEP0034900
CPT1A	Carnitine palmitoyltransferase 1α	qMmuCEP0054021
HADHA	Mitochondrial trifunctional protein, subunit α	qMmuCEP0054151
HADHB	Mitochondrial trifunctional protein, subunit β	qMmuCIP0062992

## Data Availability

All data are available upon request to the corresponding author.

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
