# Peer review of "Oral Supplementation with the Polyamine Spermidine Affects Hepatic but Not Pulmonary Lipid Metabolism in Lean but Not Obese Mice"

_nutrients, 2022, doi:10.3390/nu14204318_

Round 1

Reviewer 1 Report

In this manuscript (Ref: nutrients-1940502) with title: The polyamine spermidine affects hepatic, but not pulmonary 2 lipid metabolism in a diet-dependent manner, the authors tested oral spermidine-supplementation as intervention strategy against sucrose or fat-induced alterations of the systemic, hepatic and pulmonary lipid homeostasis. For this purpose, they gave 3mM spermidine via the drinking water to animals with different diets. And they concluded that oral spermidine-supplementation affects lipid metabolism in a diet-dependent manner, with significant reductions of body fat and weight under physiological nutrition, positive effects on weight and blood glucose under high sucrose intake, but no impact on dietary fat-related parameters.

As a general comment, I would say that this study is not completely well designed. Nonetheless, the conclusion regarding not supporting an unrestricted recommendation of oral supplementation of spermidine as an obesity intervention strategy seems to be supported by the results. Unfortunately, the results of the supplementation on the different organs systems (eg lungs) should not be considered due to huge mistakes in the experimental design that the authors themselves mention.  Even so, I have doubts that the determinations are sufficient to elucidate the mechanism under some of the metabolic changes observed. 

I outlined below some of my comments, among others that I recommend being addressed.

-First of all, the introduction is very messy, apparently there is no clear cone structure. Also, there is missing information on the status of the question. There is no info about the different diets and their effects on the metabolic parameters.

-The aim of the question seems to be quite vague, there is no primary parameter that is considered to evaluate if the intervention strategy worked systemically, in the liver of in the lungs. 

-Please, review the abbreviations, and eliminate the unnecessary ones.

- In the study is missing water consumption and pH and osmolarity of the solution. Especially the water consumption is important because with that information you know the dose of the spermidine the mice received. 

-The differences in the blood hematological analysis (figure S4) between 3 and 9 months should be discussed in depth.

-An explanation because interscapular WAT is divided by BAT should be given.

-Subcutaneous fat, brown adipose tissue, lugns and liver weight should be provided and explained in the results and discussion section.

-In intervention for reducing weight is important to measure the energy that enters but also the energy that goes out. Oxygen consumption (metabolic cage) values or thermogenesis markers should be measured. Those could be key to explain where is the energy consumed going.

-Figure 4 presents the misspelled gene abbreviations. Please, correct the names of the genes following the appropriate nomenclature (mouse MGI).

-Figure 4 and the legend does not present any type of units. 

-Mechanistically there is no explanation, nor any experiment to corroborate why spermidine produces more effects in CD than in HSD or HFD.

- To look at the direct effect of spermidine on the lungs, another experimental design should be carried out, since, as the authors explain, oral administration is not the best option, therefore the effects we see are indirect. Actually, the parameter that the authors decide to analyze does not present any alteration with the diet, therefore an improvement with spermidine could not be expected. There are other parameters that are more relevant than LD in the lungs to analyze (eg thickness of endothelium).

-The authors should put more emphasis than what appears in the introduction in explaining why they want to analyze the relationship between the benefits of spermidine with the liver and lungs and not with other tissue. Since there is only one series of qPCR and some histological images of both tissues. From my modest opinion there is no good explanation why those tissues.

Author Response

-First of all, the introduction is very messy, apparently there is no clear cone structure. Also, there is missing information on the status of the question. There is no info about the different diets and their effects on the metabolic parameters.

Answer: The introduction is subdivided into different topics. These are paragraph by paragraph: i) obesity and anti-obesity intervention strategies, ii) polyamines in general, iii) polyamines/spermidine and lipid/glucose metabolism, iv) the liver and obesity and polyamines, v) the lung and obesity and polyamines. In our opinion, these topics directly introduce the main question of the study: Does oral spermidine supplementation in combination with different diets affect systemic, hepatic or pulmonary metabolic changes? The relevance of this question is explained in more detail below.

Regarding the second point - the first paragraph of the introduction has been rephrased to add more information about previous findings regarding metabolic effects of sucrose- and fat-intake.

-The aim of the question seems to be quite vague, there is no primary parameter that is considered to evaluate if the intervention strategy worked systemically, in the liver of in the lungs.

Answer: The main question of the study was whether oral supplementation of the polyamine spermidine might be an effective intervention strategy against nutrition-related metabolic alterations. A therapeutic potential of spermidine was suggested by previous studies using genetic mutations or intraperitoneal injection. However, to use spermidine as therapeutic approach, the oral route as a physiological administration applicable for patients should be effective as well. Studies exploring oral supplementation of spermidine are scarce, and it was not addressed so far whether spermidine-related effects depend on the underlying nutrition regimen or differ between organ systems. These are important questions with regard to a possible therapeutic potential of spermidine.

The analyzed parameters in this study are key measures to assess obesity-related morbid consequences and metabolic imbalances, namely body weight, lipid deposition (in fat depots, liver and lung), circulating lipids (total, HDL- and LDL- cholesterol, triglycerides), hepatic and pulmonary expression of lipid metabolism-related enzymes as well as glucose homeostasis (oral glucose tolerance testing, fasting glucose and insulin concentrations). In our opinion these parameter variety provides a comprehensive evaluation of spermidine-induced metabolic effects at various levels and in different organ systems. The last paragraph of the introduction has been rephrased to emphasize these points in a more direct way.

-Please, review the abbreviations, and eliminate the unnecessary ones.

Answer: Thank you for this comment, this has been done.

- In the study is missing water consumption and pH and osmolarity of the solution. Especially the water consumption is important because with that information you know the dose of the spermidine the mice received.

Answer: These are valid points. Drinking bottles were regularly weighed, but the actual intake of water could not be differentiated from fluid spillage. However, in previous studies we demonstrated that spermidine administration as done in the current study resulted in a significant increases of spermidine levels (Eisenberg et al. Nat Med. 2016; Ahrendt et al. Am J Physiol Lung Cell Mol Physiol. 2020). The pH and osmolality of the spermidine containing drinking water (pH 7.4, 16 mosm/kg) has been added to Material and Methods.

-The differences in the blood hematological analysis (figure S4) between 3 and 9 months should be discussed in depth.

Answer: Unfortunetaly, a figure S4 does not exist in the manuscript.

-An explanation because interscapular WAT is divided by BAT should be given.

Answer: Thank you for your comment. The labelling “WAT/BAT” was meant to emphasize that the interscapular fat depot is composed of white adipose tissue and brown adipose tissue. The labelling has been rephrased to “WAT+BAT” to clarify that.

-Subcutaneous fat, brown adipose tissue, lugns and liver weight should be provided and explained in the results and discussion section.

Answer: In this study we focused on 3 major, delimited fat depots that can be isolated in a comparable way for all body weight conditions. The interscapular fat depot is one of the anterior subcutaneous depots that contain brown adipose tissue (Chusyd et al. Front Nutr. 2016), the corresponding weights are given in Fig. 1 and are discussed in the manuscript. Lung volumes and liver weights have been added to Fig. 6 (former Fig. 5).

-In intervention for reducing weight is important to measure the energy that enters but also the energy that goes out. Oxygen consumption (metabolic cage) values or thermogenesis markers should be measured. Those could be key to explain where is the energy consumed going.

Answer: The authors agree, that measurement of energy expenditure would have been an interesting additional parameter, that was not possible due to lack of experimental equipment. This point has been added as a limitation of the study to the discussion section.

-Figure 4 presents the misspelled gene abbreviations. Please, correct the names of the genes following the appropriate nomenclature (mouse MGI).

Answer: This has been corrected.

-Figure 4 and the legend does not present any type of units.

Answer: Expression data is presented as fold expression values as explained in Material and Methods. However, this information has been added to the figure legend of new Fig. 5 (former Fig. 4) for clarification.

-Mechanistically there is no explanation, nor any experiment to corroborate why spermidine produces more effects in CD than in HSD or HFD.

Answer: This study was designed as an explorative investigation to test whether spermidine supplementation might be a treatment option for overweight-related metabolic changes. Our comprehensive analysis includes different dietary regimens (balanced diet, high carbohydrate intake, high fat intake) and a variety of metabolic key measures in different organ systems. By this approach we were able to show that spermidine affects the lipid metabolism, but depending on the nutritional status and organ system. This warrants future studies focusing on further elucidation of the molecular mechanisms underlying these findings. 

- To look at the direct effect of spermidine on the lungs, another experimental design should be carried out, since, as the authors explain, oral administration is not the best option, therefore the effects we see are indirect. Actually, the parameter that the authors decide to analyze does not present any alteration with the diet, therefore an improvement with spermidine could not be expected. There are other parameters that are more relevant than LD in the lungs to analyze (eg thickness of endothelium).

Answer: Thank you for raising this point, although we partly disagree. In our opinion, oral supplementation of spermidine as physiologic route would be the best option if spermidine should be used as a therapeutic intervention strategy in humans, in contrast to intraperitoneal injections or genetic alterations as analyzed in other reports. Thus, it is reasonable to focus on this administration route. Moreover, oral spermidine administration was shown before to result in increased pulmonary spermidine concentrations and alleviation of obesity-related capillary remodeling (Ahrendt et al. Am J Physiol Lung Cell Mol Physiol. 2020), confirming the bioavailability of dietary spermidine to the lung. Since this study focuses on the lipid metabolism, pulmonary lipid accumulation and expression of lipid metabolism-related enzymes were analyzed and directly compared to other organs like the liver. However, we agree with the suggestion of the reviewer that e.g. analysis of lung structural parameters would be very interesting and should be addressed in follow-up studies.

-The authors should put more emphasis than what appears in the introduction in explaining why they want to analyze the relationship between the benefits of spermidine with the liver and lungs and not with other tissue. Since there is only one series of qPCR and some histological images of both tissues. From my modest opinion there is no good explanation why those tissues.

Answer: Thank you for this comment. It is more and more acknowledged that metabolic disorders affect many organs and cause a multimorbid state in obese and diabetic individuals (Kivimäki et al. Lancet Diabetes Endocrinol. 2022). Single studies indicate that liver and lung pathologies are linked to each other. Patients with liver diseases are at higher risk for ARDS, and a case study demonstrated that liver transplantation was efficient to resolve ARDS, implying that a healthy liver protects the lung from injury (Herrero et al. Intensive Care Med Exp. 2020).  Hepatic steatosis accompanies alveolar proteinosis, indicating close links between the pulmonary and the hepatic lipid metabolism (Hunt et al. Am J Respir Cell Mol Biol. 2017). These points have been added to the introduction. 

Reviewer 2 Report

This is an interesting study investigating the positive association between oral spermidine-supplementation and lipid metabolism in a diet-dependent manner.

The paper is very interesting and a more detailed analysis of the pulmonary cellular alterations, as anticipated by the authors, would be certainly very helpful

Minor revision

-The authors should better clarify the aim of the paper for the purpose of its hypothetical use in human pathologies such as obesity and metabolic syndrome

- The total number of animals used is missing. Authors should also report the number of animals used for each group

-A flowchart with the indications of the evaluations over the different weeks would help a lot

Author Response

Reviewer 2

-The authors should better clarify the aim of the paper for the purpose of its hypothetical use in human pathologies such as obesity and metabolic syndrome

Answer: Thank you for your input. We rephrased the aim of the study at the end of the introduction for clarification. 

- The total number of animals used is missing. Authors should also report the number of animals used for each group

Answer: The animal numbers for shown mean values have been added to the figure legends. In the dot plots every symbol represents one animal, thus directly stating the number of analyzed animals.

-A flowchart with the indications of the evaluations over the different weeks would help a lot

Answer: Thank you for this comment. A scheme explaining the experimental design has been added to the Material & Methods section (new figure 1).

Reviewer 3 Report

In this manuscript are describing the spermidine and spermine effect under different dietary conditions. I found in this study an excellent experimental design addressing the main hot topics to aim by functional supplements for diet-induced hepatic and lung disfunctions. Authors are presenting a clarifying introduction to understand the results and discussion. The results are clearly displayed. The discussion is widely and richly carried out highlighting limitations and discrepancies with other studies of reference. However, before its full acceptance, I would like to comment and suggest certain aspects:

Lines 39-41: A suitable reference is needed.

Lines 50-52: Authors mentioned that “Degradation and secretion of spermidine and spermine requires acetyl-CoA-dependent acetylation by SSAT”. In other words, depletion of acetyl-CoA is the mechanism by which spermidine and spermine perform their lipogenic and cholesterogenic functions. Although, authors have quantified different genes that might indicated an increased activity of SSAT, they did not quantify either SSAT expression or the active protein levels. Why? This should be better explain in the discussion section.

Table 1: Acronyms in the headlines should be defined in the table legend.

Line 146: “of 20 cmH2O,” a space is needed

Author Response

Reviewer 3

Lines 39-41: A suitable reference is needed.

Answer: The reference “Choksomngam et al. The Metabolic Role of Spermidine in Obesity: Evidence from Cells to Community. Obes. Res. Clin. Pract. 2021” has been added.

Lines 50-52: Authors mentioned that “Degradation and secretion of spermidine and spermine requires acetyl-CoA-dependent acetylation by SSAT”. In other words, depletion of acetyl-CoA is the mechanism by which spermidine and spermine perform their lipogenic and cholesterogenic functions. Although, authors have quantified different genes that might indicated an increased activity of SSAT, they did not quantify either SSAT expression or the active protein levels. Why? This should be better explain in the discussion section.

Answer: This is an interesting point. Although depletion of acetyl-CoA is the main effect induced by SSAT overexpression (Jell et al. J. Biol. Chem. 2007; Kramer et al. J. Biol. Chem. 2008; Liu et al. Amino Acids 2014; Pirinen et al. Mol. Cell. Biol. 2007) this not necessarily holds true for spermidine supplementation. It is also conceivable that spermidine-induced autophagy confers or contributes to the observed effects, like it was shown in aged organisms (Eisenberg et al. Nat. Cell Biol. 2009; Eisenberg et al. Nat. Med. 2016). The current study was designed as an explorative investigation to find out whether or not oral spermidine supplementation influence fat- and/or sugar-related metabolic changes. Our results warrant future studies focusing on the molecular mechanisms underlying the findings reported here.

Table 1: Acronyms in the headlines should be defined in the table legend.

Answer: This has been added.

Line 146: “of 20 cmH2O,” a space is needed

Answer: This has been corrected.

Round 2

Reviewer 1 Report

I believe that the article in its current state does not add anything new to the field. The authors have studied the oral effect of spermidine, an effect that, as they themselves comment, has already been described previously. The aim of the study is not clear, it is very exploratory. The authors suggest that its added value lies in the fact that they wanted to study it depending on the underlying nutrition regimen. In the absence of major dietary changes: HFD increases body weight, insulin resistance, and blood and liver lipids; HSF, on the other hand, only increases lipids in blood and liver; making it very difficult to know if there is a beneficial effect of spermidine. Many of the parameters altered by the diet are slightly improved but without reaching the values of the CD diet. If the aim is that spermidine helps to lose weight, it does not succeed, if the intention is to improve insulin sensitivity, it does not succeed, if the intention is to improve the lipid profile, it does improve slightly, but only in the bloodstream no its deposition in the liver. These results make a follow-up study explaining the mechanism very difficult. In such a case, the ideal would always be to compare it with caloric restriction.

I understand that the authors wanted to publish it in its current state since its experimental design lacks a group with caloric restriction, a treatment which spermidine wants to improve or match, which makes its interpretation difficult. A result is a result, and therefore even if it is not positive, or designed with some flaws it should be published. I would recommend making it clearer in the conclusions that the dose and dosage form has not improved the metabolic profile of the animals on the diets. And try to explain in more detail some theories why this has happened. A more appropriate title referring to the absence of effects with HFD could be a possibility to be evaluated.

Author Response

Dear reviewer,

Thank you for your comments. Your suggestions have been addressed as follows:

The title now reads: "Oral supplementation with the polyamine spermidine affects hepatic, but not pulmonary lipid metabolism in lean, but not obese mice".

We added a new paragraph to the discussion section that highlights differences to other studies that reported divergent effects of spermidine supplementation in other experimental setups. 

Moreover, we rephrased the conclusion, it now reads: "Taken together, our data show that oral administration of 3 mM spermidine is able to influence the lipid metabolism, but its eventual effectiveness is dependent on the nutritional status of the individual. In control diet-fed, lean mice, spermidine reduced body weight and lipid contents in adipose tissues and the liver. Upon high sucrose intake, it still diminished body weight and blood glucose. However,  spermidine-supplementation of HFD-fed obese mice exerted no effects neither on body weight nor on lipid deposition. This indicates that spermidine-supplementation is effective as CRM, but only under specific diet-related conditions. The dose and dosage form of spermidine used here did not improve the metabolic profile of diet-induced obese mice, thus, this study does not support an unrestricted recommendation of spermidine as obesity intervention strategy."

We look forward to hearing from you regarding our revision.